# A comparative study of the capacity of mesenchymal stromal cell lines to form spheroids

Margaux Deynoux[1]☯, Nicola Sunter[1]☯, Elfi Ducrocq[1], Hassan Dakik[1], Roseline Guibon[2,3], Julien Burlaud-Gaillard[4,5], Lucie Brisson[3], Florence Rouleux-Bonnin[1], Louis-Romée le Nail[6], Olivier Hérault[1,7], Jorge Domenech[1,7], Philippe Roingeard[4,5], Gaëlle Fromont[2,3], Frédéric Mazurier[1]*

1 EA 7501 GICC, CNRS ERL 7001 LNOx, Université de Tours, Tours, France, 2 Anatomie et cytologie pathologique, CHRU de Tours, Tours, France, 3 INSERM UMR1069, Nutrition, Croissance et Cancer, Université de Tours, Tours, France, 4 Plateforme IBiSA de Microscopie Electronique, Université et CHRU de Tours, Tours, France, 5 INSERM U1259 MAVIVH, Université et CHRU de Tours, Tours, France, 6 Service de Chirurgie orthopédique, CHU Tours, Tours, France, 7 Service d'hématologie biologique, CHRU de Tours, Tours, France

☯ These authors contributed equally to this work.
* frederic.mazurier@inserm.fr

**Data Availability Statement:** All relevant data are within the paper and its Supporting Information files.

## Abstract

Mesenchymal stem cells (MSC)-spheroid models favor maintenance of stemness, *ex vivo* expansion and transplantation efficacy. Spheroids may also be considered as useful surrogate models of the hematopoietic niche. However, accessibility to primary cells, from bone marrow (BM) or adipose tissues, may limit their experimental use and the lack of consistency in methods to form spheroids may affect data interpretation. In this study, we aimed to create a simple model by examining the ability of cell lines, from human (HS-27a and HS-5) and murine (MS-5) BM origins, to form spheroids, compared to primary human MSCs (hMSCs). Our protocol efficiently allowed the spheroid formation from all cell types within 24 hours. Whilst hMSC-spheroids began to shrink after 24 hours, the size of spheroids from cell lines remained constant during three weeks. The difference was partially explained by the balance between proliferation and cell death, which could be triggered by hypoxia and induced oxidative stress. Our results demonstrate that, like hMSCs, MSC cell lines make reproductible spheroids that are easily handled. Thus, this model could help in understanding mechanisms involved in MSC functions and may provide a simple model by which to study cell interactions in the BM niche.

## Introduction

Over the last two decades, extensive studies have attempted to characterize mesenchymal stem cell (MSC). Initially described in the bone marrow (BM), MSCs were later found in almost all adult and fetal tissues [1]. Their classification rapidly suffered from a lack of clear phenotypical definition. Therefore, in 2006, the International Society for Cellular Therapy (ISCT) defined

**Funding:** The authors acknowledge the Ministry of Research (MD), the ARC Foundation (MD, HD), the French Society of Hematology (MD), the "Ligue contre le Cancer (NS)", and the Lebanese south governate (HD) for their funding. This work was supported by the French Committees of the "Ligue Contre le Cancer Grand-Ouest" [16 (Charente), 36 (Indre), 37 (Indre-et-Loire), 41 (Loire et Cher), and 86 (Vendée)] and the Région Centre Val de Loire (FM). The funders had no role in study design, data collection and analysis, decision to publish, or preparation of the manuscript.

**Competing interests:** The authors have declared that no competing interests exist.

MSCs according to three minimal criteria: adherence to plastic, specific cell surface markers and multipotent potential. Indeed, MSCs are classically described as stem cells that are able to differentiate into osteoblasts, adipocytes and chondroblasts [2], making them an attractive source of cells in regenerative medicine. Subsequent studies have also established their ability to differentiate into cardiomyocytes [3], neurons [4], epithelial cells [5] and hepatocytes [6]. The discovery of the multiple functions of MSC, such as those involved in the anti-inflammatory response [7] and in injury repair [8,9] confirmed them as promising cellular tools in regenerative medicine.

Furthermore, MSCs represent a key component of the BM microenvironment supporting normal hematopoiesis through the regulation of stem cell renewal and differentiation processes, but also fueling malignant cells and protecting them from therapeutic agents [10]. As such, primary MSCs have often been used as feeder layers in long-term co-culture of hematopoietic cells *in vitro* in preclinical studies [11]. With the aim of standardization, the murine MS-5 cell line became a standard for both normal or malignant hematopoietic cell culture [12]. This robust co-culture model has been widely used and has contributed to the characterization of hematopoietic stem cells (HSC) [11]. This two-dimensional (2D) system, while closer to BM physiology than the culture of hematopoietic cells alone, still lacks the three-dimensional (3D) complexity of the BM niche. Thus, although widely used, it is certainly not sufficiently consistent at predicting *in vivo* responses [13]. Therefore, a 3D system might be a better alternative to mimic the BM microenvironment.

Critically, the culture leads to rapid loss of MSC pluripotency and supportive functions. Therefore, a wide range of techniques to form 3D MSCs aggregates, from the simplest spheroids to the more complex matrix-based structures, have been proposed [14]. Studies of spheroids, also called mesenspheres, were mostly dedicated to the examination of MSC stemness and differentiation abilities, such as osteogenesis, in order to improve their *in vitro* expansion and transplantation efficacy in regenerative medicine [15,16]. Furthermore, this model has also been tested as a surrogate niche for hematopoietic cells [17–23]. Spheroids take advantage of the ability of MSCs to self-aggregate, which is improved by using various approaches such as low adhesion plates, natural and artificial (centrifugation) gravity, cell matrix or more complex scaffolds [13,14,24,25]. Classically, studies have used human primary MSCs, from BM, cord blood and lipoaspirate, or rodent sources [15,26].

Although immortalized MSCs, or well characterized cell lines, could bypass the lack of primary cells and avoid the variability involved with use of primary human MSCs (hMSCs) samples, they are rarely employed to make spheroids [27,28]. Cell lines would also allow better standardization of the spheroid formation protocol. In this study, we examined the spheroid-forming capacity of two human cell lines (HS-27a and HS-5) and the currently used murine MS-5 cell line, in comparison with primary hMSCs. We defined a simple and fast method using standard matrix to form spheroids and characterized them in terms of physical features, cell proliferation and death.

## Materials and methods

### Cell culture and reagents

The murine MS-5 bone marrow (BM) stromal cell line was kindly provided by Mori KJ (Niigata University, Japan) [29]. HS-27a and HS-5 human BM stromal cell lines were purchased from American Type Culture Collection (CRL-2496 and CRL-11882, respectively). Primary BM hMSCs were obtained by iliac crest aspiration from healthy donors (without hematological disorders) undergoing orthopedic surgery at the University Hospital of Tours, after informed consent, for cell banking according to the Declaration of Helsinki, as approved by the French

Ministry of Education and Research (authorization number No. DC-2008-308). Samples from BM aspirates were diluted in MEM Alpha (Life Technologies, Villebon-sur-Yvette, France) and filtered through a cell-strainer prior to centrifugation (350 x g, 10 min). Cells were resuspended and seeded at $10^5$ to $2.10^5$ cells/cm$^2$ in MEM Alpha supplemented with 100 U/mL penicillin and 100 µg/mL streptomycin (both from Life Technologies) and 1 ng/mL of recombinant human FGF basic (FGF-2, R&D Systems, Abingdon, United Kingdom). Medium was changed twice a week until cells reached confluency. In experiments, primary hMSCs were used at passage 2. HS-27a and HS-5 cell lines were cultured in RPMI 1640 (Life Technologies) and MS-5 in MEM Alpha. All media were supplemented with 10% heat-inactivated fetal bovine serum (FBS), 2 mM L-glutamine (Life Technologies), 100 U/mL penicillin and 100 µg/mL streptomycin. Cells were maintained in a saturated humidified atmosphere at 37˚C and 5% $CO_2$.

## Spheroids formation

For one spheroid, 30,000 cells were cultured in 100 µL of medium, supplemented by 0.25% to 1% of either Methocult$^{TM}$ H4100 or SF H4236 (StemCell, Grenoble, France), and seeded in U-bottomed 96-well plate (Sarstedt, Marnay, France). Both media contains methylcellulose in IMDM, but SF H4236 is supplemented with bovine serum albumin, recombinant human insulin, human transferrin (iron-saturated), 2-Mercaptoethanol and unknown supplements as described by the manufacturer. The medium was the same as that of the normal culture for each cell line but supplemented with heat inactivated FBS to reach 15%. At days as detailed, microscopic analysis was performed using a Leica DMIL microscope (Leica, Nanterre, France), coupled to a DXM1200F camera (Nikon, Champigny-sur-Marne, France). To determine the number of cells in each spheroid over time, only wells with a unique, fully-formed spheroid were selected. Twelve spheroids per experiment were pooled and dissociated with 2 mg/mL collagenase 1A (Sigma-Aldrich, Saint-Quentin-Fallavier, France), 10 min at 37˚C, with agitation every two minutes, and then counted by the trypan blue exclusion assay.

## Time-lapse video

Automatic acquisitions were performed on a Nikon Eclipse TI-S microscope, coupled to a DS Qi2 camera (Nikon). The system includes a cage incubator (Okolab, Pozzuoli, NA, Italy) controlling temperature and level of $CO_2$. Analyses were performed using both NIS Element BR (Nikon) and Fiji/ImageJ softwares.

## Scanning electron microscopy

Spheroids were fixed by incubation for 24 h in 4% paraformaldehyde, 1% glutaraldehyde in 0.1 M phosphate buffer (pH 7.2). Samples were then washed in phosphate-buffered saline (PBS) and post-fixed by incubation with 2% osmium tetroxide for 1 h. Spheroids were then fully dehydrated in a graded series of ethanol solutions, and dried in hexamethyldisilazane (HMDS, Sigma-Aldrich). Finally, samples were coated with 40 Å platinum, using a PECS 682 apparatus (Gatan, Evry, France), before observation under an Ultra plus FEG-SEM scanning electron microscope (Zeiss, Marly-le-Roi, France).

## Transmission electron microscopy

Spheroids were fixed by incubation for 24 h in 4% paraformaldehyde, 1% glutaraldehyde in 0.1 M phosphate buffer (pH 7.2). Samples were then washed in phosphate-buffered saline (PBS) and post-fixed by incubation with 2% osmium tetroxide for 1 h. Spheroids were then fully

dehydrated in a graded series of ethanol solutions and propylene oxide. Impregnation step was performed with a mixture of (1:1) propylene oxide/Epon resin, and then left overnight in pure resin. Samples were then embedded in Epon resin, which was allowed to polymerize for 48 h at 60°C. Ultra-thin sections (90 nm) were obtained with an EM UC7 ultramicrotome (Leica). Sections were stained with 5% uranyl acetate (Agar Scientific, Stansted, United Kingdom), 5% lead citrate (Sigma-Aldrich) and observations were made with a transmission electron microscope (Jeol, JEM 1011, Croissy-sur-Seine, France).

### Immunohistochemistry

At least five spheroids per condition were pooled, fixed in formalin, embedded in paraffin and cut in 3–4 μm sections on Superfrost Plus slides. Slides were deparaffinized, rehydrated and heated in citrate buffer pH 6 for antigenic retrieval. After blocking for endogenous peroxidase with 3% hydrogen peroxide, the primary antibodies were incubated. The panel of primary antibodies included anti-HIF-1α (Abcam ab51608, Paris, France) (dilution 1/200, incubation 1 h), anti-VEGF-A (Abcam ab1316, dilution 1/200, incubation 1 h), anti-HO-1 (Abcam ab52947, dilution 1/1 000, incubation 1 h), anti-CA-IX (Novocastra clone TH22, Nanterre, France) (dilution 1/100, incubation 20 min), anti-Ki-67 (DakoCytomation clone 39–9, Glostrup, Denmark) (dilution 1/50, incubation 30 min), anti-caspase 3 (Novocostra clone JHM62, Nanterre, France) (dilution 1/100, incubation 1 h) and anti-LC3B (Novus Biological NB 600–1384, Cambridge, UK) (dilution 1/200, incubation 1 h). Immunohistochemistry was performed with either the automated BenchMark XT slide stainer (Ventana Medical System Inc.) using OptiView Detection Kit (Ventana Medical System Inc.) (for CA-IX and Ki-67), or manually using the streptavidin-biotin-peroxidase method with diaminobenzidine as the chromogen (Kit LSAB, DakoCytomation). Slides were finally counterstained with haematoxylin. Negative controls were obtained after omission of the primary antibody or incubation with a non-specific antibody.

### Quantitative real-time PCR

Total RNAs were extracted using TRIzol reagent (Life Technologies) and reverse transcription was performed with the SuperScript$^{TM}$ VILO$^{TM}$ cDNA Synthesis Kit (Invitrogen, Villebon-sur-Yvette, France), both according to the manufacturer's procedures. qRT-PCR was performed on a LightCycler® 480 (Roche, Switzerland) with the LightCycler® 480 Probes Master (Roche). *GAPDH*, *ACTB*, *RPL13A* and *EF1A* genes were used as endogenous genes for normalization. Primer sequences (S1 Table) were designed with the ProbeFinder software (Roche), and all reactions were run in triplicate.

### Cell cycle analysis

Spheroids were dissociated with 2 mg/mL collagenase 1A (Sigma-Aldrich), 10 min, at 37°C, with agitation every two minutes. Cells were fixed with 2% paraformaldehyde/0.03% saponin for 15 min at room temperature (RT), and washed three times for 5 min with 10% FBS/0.03% saponin. Cells were then stained with 7-Aminoactinomycin D (7-AAD, Sigma-Aldrich) and an AF488-conjugated anti-KI-67 antibody (BD Biosciences, Le Pont de Claix, France) or the AF488-conjugated IgG$_1$ isotype control (BD Biosciences). Experiments were performed on an Accuri$^{TM}$ C6 flow cytometer (BD Biosciences) and data were analyzed with the FlowJo V10.4.1 software (Tree Star Inc.).

## Statistical analysis

All statistical analyses were performed using R software. Since our sampling never exceeds n = 6, we used nonparametric tests. The Mann-Whitney test was used to compare two conditions and Kruskal-Wallis for multiple comparisons, followed by a Dunn's *post hoc* test. The threshold for significance was set up to a p-value of 0.05.

## Results

### Establishment of primary hMSC-spheroids by cell aggregation method

From the different methods to form MSC-spheroids, we followed an approach based on cell aggregation in methylcellulose-based medium [27]. To establish a protocol that is simple, reproducible and compatible with hematopoietic cell culture, two commercial methylcelluloses (MethoCult H4100 and SF H4236) developed for hematopoietic progenitors assays were tested (Fig 1A). A range from 0.01 to 1% of methylcellulose has been previously used [27,30–33], so we tested three different concentrations (0.25, 0.5 and 1%). We also tested the hanging drop technique [31,33–35] and the previously described U-bottomed 96-well plates methods [27,30,32,33,36]. Both techniques worked well for primary hMSCs but the second was more appropriate for further analyses since the handling is easier and the volume of medium higher, which could prevent starvation and dehydration in long-term cultures. The SF H4236 methylcellulose at a concentration of 0.5% was adopted because it generated only one spheroid in most of the wells with lower condensation aspect for primary hMSCs (Fig 1B). Under these culture conditions, MSCs were able to form spheroids rapidly, in as little as five hours of culture (S1 Video), which is consistent with previous studies [27,32,37].

### Formation of spheroids from MSC lines

The spheroid-forming capacity was followed for two human cell lines, HS-27a and HS-5, and compared to that of primary hMSCs. The two cell lines were obtained by immortalization of hMSCs from the same BM sample with the papilloma virus E6/E7 genes [38,39]. HS-27a cells support hematopoietic stem cell maintenance (self-renewal, formation of cobblestone areas), whereas HS-5 cells mainly sustain proliferation and differentiation [38–40]. Like primary hMSCs, they both retained the ability to form spheres but required about 10 hours to make rounded spheroids (S2 and S3 Videos). Although cells of various origins formed spheroids of equivalent sizes (about 300 μm of diameter) after 24 hours, primary hMSC-spheroids rapidly condensed and reached half of their initial perimeter after 14 days of culture (Fig 2A and 2B). In contrast to primary hMSCs, the perimeter of spheroids resulting from both cell lines remained constant during three weeks. Knowing that primary hMSCs and cell lines may differ in their growth properties, we used the murine MS-5 cell line that has contact inhibition [29]. This cell line was able to quickly form spheroids similarly to the other cell lines (S4 Video). It is noteworthy that MS-5 cells initially formed a flat multilayer disk of cells prior to contracting into spheres. Similarly to the spheroids from human cell lines, spheroids from MS-5 cells kept the same size over time (Fig 2A and 2B). This suggests that shrinking might be an intrinsic property or extracellular matrix (ECM) composition of primary cells rather than related to cell proliferation control. We thus examined whether the difference in the size maintenance between various MSCs might be attributed to the cell number per spheroid. In order to quantify the viable cells, spheroids were dissociated at different timepoints after seeding. In accordance with the decrease in circumference, the number of cells per spheroid from primary hMSCs dramatically dropped within seven days (Fig 2C), in agreement with other studies [31,35]. Remarkably, although keeping the same size, HS-27a-spheroids, as well as the MS-5

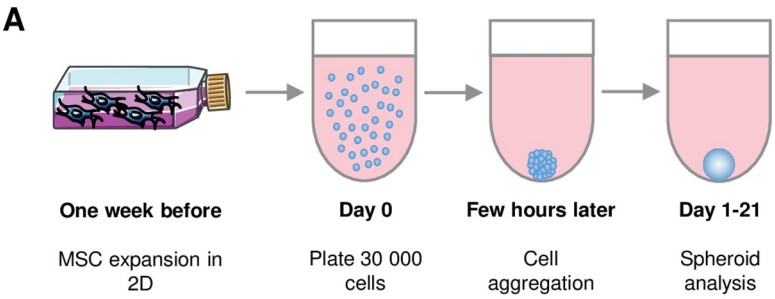

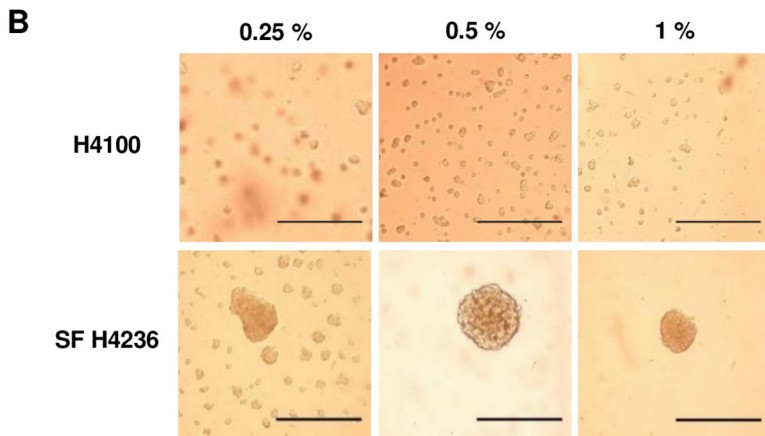

**Fig 1. Spheroids formation from primary hMSCs.** (A) Schematic representation of experimental plan. (B) 30,000 primary hMSCs per well were seeded into U-bottomed 96-well in medium containing 0.25%, 0.5% or 1% of methylcellulose (Methocult™ H4100 or SF H4236). Microscopy analysis was performed after 24 h (scale bars = 500 μm).

ones, had lost viable cells similarly to primary hMSCs (Fig 2C). In contrast, HS-5-spheroids had less obvious decrease in cell number with time (Fig 2C). Overall, the size reduction does not seem to be strictly attributable to reduced cell number in spheroids and could be possibly attributed to other factors such as the ECM composition.

## Electron microscopy observation of the MSC-spheroids

Scanning electron microscopy (SEM) confirmed the shrinking of primary hMSC-spheroids (Fig 3A and S1A Fig). SEM also revealed, at higher magnification, that spheroids from primary hMSCs are highly cohesive, showing tight intercellular connections forming a flat surface, whereas HS-27a- and HS-5-spheroids, and to a lesser extend MS5-spheroids, exhibited more rounded cells at their surface (Fig 3B and S1B Fig). This phenomenon intensified over time and may explain the size reduction of hMSC-spheroids compared to the cell lines. Absence of ECM is not involved, since ECM deposition is visible for all cell types (Fig 3B). From day 7 for cell lines and day 14 for primaries, spheroid structure began to change, showing loss of cell-cell adhesion, and cell death at the surface.

Further analysis by transmission electron microscopy (TEM) was performed to investigate the ultrastructure of the cells within the spheroids. Between day 1 and day 7, cells showed the swelling of the cell cytoplasm and the presence of an increasing number of necrotic cells, thus suggesting particularly rapid induction of cell death for primary hMSCs compared to cell lines

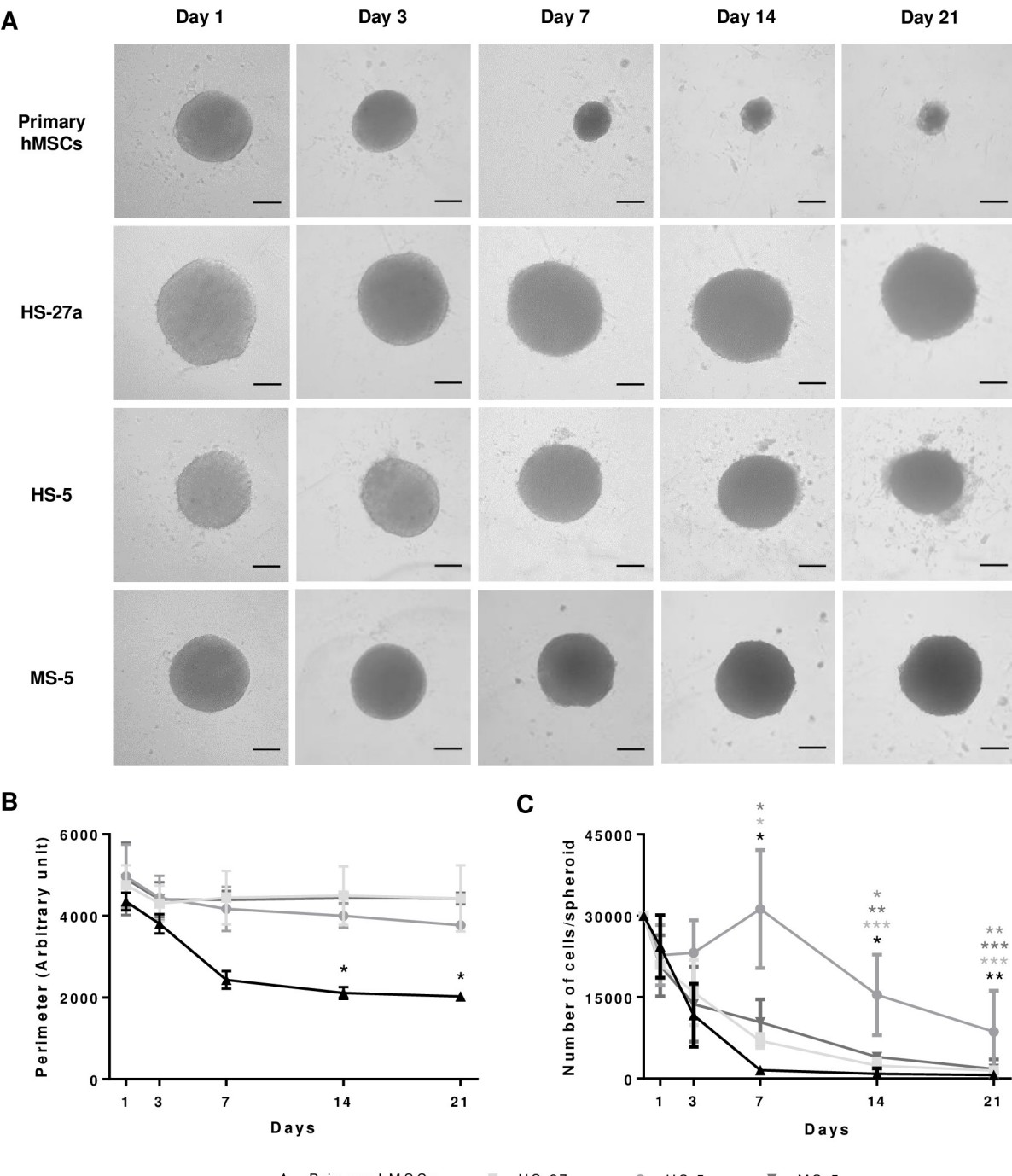

**Fig 2. Follow up of the spheroids from various MSCs.** (A) Microscopy analysis of primary hMSC-, HS-27a-, HS-5- and MS-5-spheroids over 21 days in culture (scale bars = 100 μm). (B) Perimeter was measured with an arbitrary unit; each experiment is the mean of at least 10 spheroids from n = 3 experiments. Data are shown as mean ± SD; * compared to day 1; * p ≤ 0.01. (C) Number of living cells per spheroid over 21 days in culture (primary hMCSs and MS-5 n = 3; HS-27a and HS-5 n = 4). Data are shown as mean ± SD; *, **, *** compared to day 0; * p ≤ 0.05; ** p ≤ 0.01; *** p ≤ 0.001.

(S2A–S2L Fig). Strong induction of autophagy, indicative of cell stress, was demonstrated by appearance of numerous cytoplasmic vacuoles and autophagosomes for primary hMSCs, as soon as day 1 (S2A Fig), and could certainly explain the shrinking. Remarkably, despite a

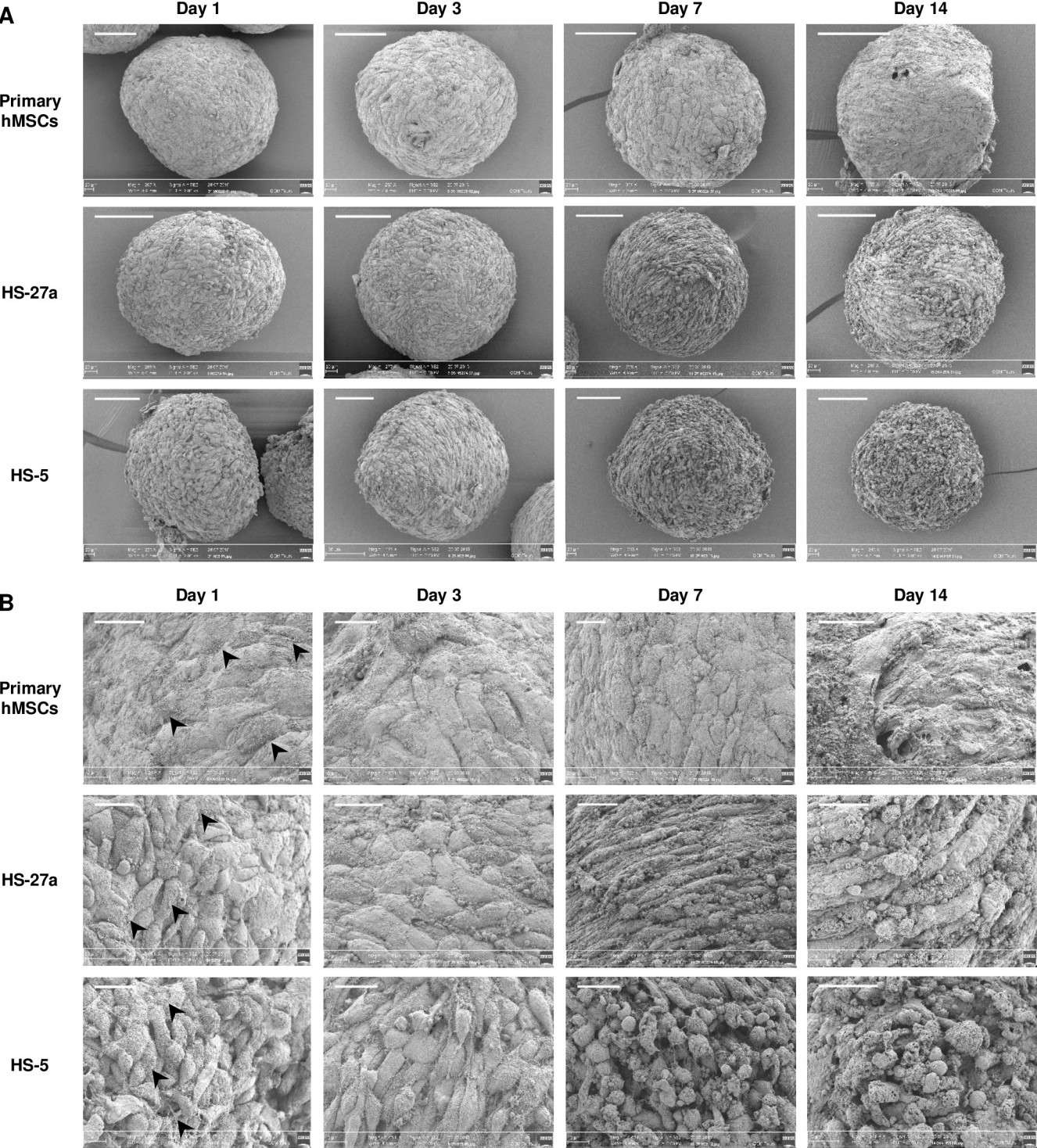

**Fig 3. Scanning electron microscopy (SEM) observation of MSC-spheroids.** (A) Spheroids from primary hMSCs are round with a smooth surface and show progressive shrinking. HS-27a- and HS-5-spheroids are more irregular and granular. (B) Higher magnification show that cells are cohesive at the surface of primary hMSC-spheroids and more chaotic with distinguishable cells of different shapes for human cell lines. ECM deposition (arrow heads) is visible on all spheroids. Scale bars = 100 μm (A) and 20 μm (B).

higher apparent loss of cell adhesion by TEM, induction of cell death (necrosis) appeared delayed for cell lines. In addition, although the number of cytoplasmic vacuoles increased over time, no autophagosomes were noted in cell lines. Consistent with this observation, although LC3B expression can be triggered in HS-27a cells (S3 Fig), its subcellular expression remains uniformly cytoplasmic rather than as dot-like staining patterns, which may thus indicate a block in the autophagic mechanism as previously described [41,42].

## Cell death and proliferation of the MSC-spheroids

To explain why spheroids showed decreased cell number over time, we proposed an imbalance between cell death and cell proliferation. Cell death has already been noted by electronic microscopy observation though autophagy and cell lysis. Apoptosis and cell cycle were first determined by flow cytometry using 7-AAD/Ki-67 staining (Fig 4A). Increasing sub-$G_0/G_1$ cell population revealed a strong induction of necrosis and/or late apoptosis after 14 days in hMSC-spheroids, whereas none or moderate cell death was observed for the two human cell lines (Fig 4B). In accordance with these data, caspase-3 staining showed few apoptotic cells until day 7 for primary hMSCs and HS-5 cells (Fig 4C). In contrast, apoptosis in HS-27a-spheroids was observed as soon as day 1, and increased with time, which is consistent with earlier detection of death cells by flow cytometry (Fig 4B). Regarding the proliferation, although harvested at the same confluency, primary hMSCs appeared already much more quiescent than HS-27a or HS-5 cells at day 0 (Fig 4D). A significant proportion of cells remained proliferating in spheroids until day 3 for HS-27a and day 7 for HS-5 cells. Remarkably, while closer to HS-27a cells in terms of perimeter and number of cells, MS-5 cells had a massive increase in cell death and almost no proliferation (S1C and S1D Fig), similarly to primary hMSCs. Ki-67 detection by immunochemistry, in primary hMSCs and human cell lines, revealed homogeneous staining at day 1 indicating proliferation in the whole spheroid (Fig 4E) in agreement with a previous study [43]. Staining confirmed a lower proliferation rate of primary hMSCs compared to cell lines and a rapid proliferation arrest with only few Ki-67-positive cells remaining at the periphery of the spheroid at day 3. A progressive decrease in proliferation for the human cell lines supported the results obtained by flow cytometry. Remarkably, decreased proliferation appears in the entire spheroid and is not restricted to in-depth localizations. These data showed that spheroids are characterized by imbalance between cell death and proliferation, which may explain the highest loss of cells over time.

## Hypoxia and oxidative stress in MSC-spheroids

Like in tumor spheres [44–46], the appearance of an oxygen gradient and hypoxia in MSC-spheroids has been demonstrated [47,48]. Carbonic anhydrase IX (CA-IX), a mediator of hypoxia-induced stress response, is commonly used as marker in tumors [49]. Increased CA-IX has been observed in MSC-spheroids, particularly for HS-27a cells (Fig 5A). The pro-survival adaptation to hypoxia occurs mainly through the stabilization of the hypoxia-inducible factors (HIFs). HIFs are key regulators of multiple cell processes, including cell cycle, metabolism, pH control and autophagy. Increasing expression of HIF-1α protein expression has been observed in spheroids over time, as well as at the mRNA level (Fig 5B). Finally, we examined the expression of *VEGFA*, a typical HIF transcriptionally regulated gene [50]. Its expression in hMSC- and HS-27a-spheroids was already elevated at day 1, but strongly increased at both protein and mRNA levels over time (Fig 5C).

In certain circumstances, very low level of oxygen (anoxia) or long exposure to hypoxia may provoke DNA damage and oxidative stress that trigger apoptosis [44,46]. Besides hypoxia appearance in spheroids, cell aggregation may also stress the cells by itself and increase reactive

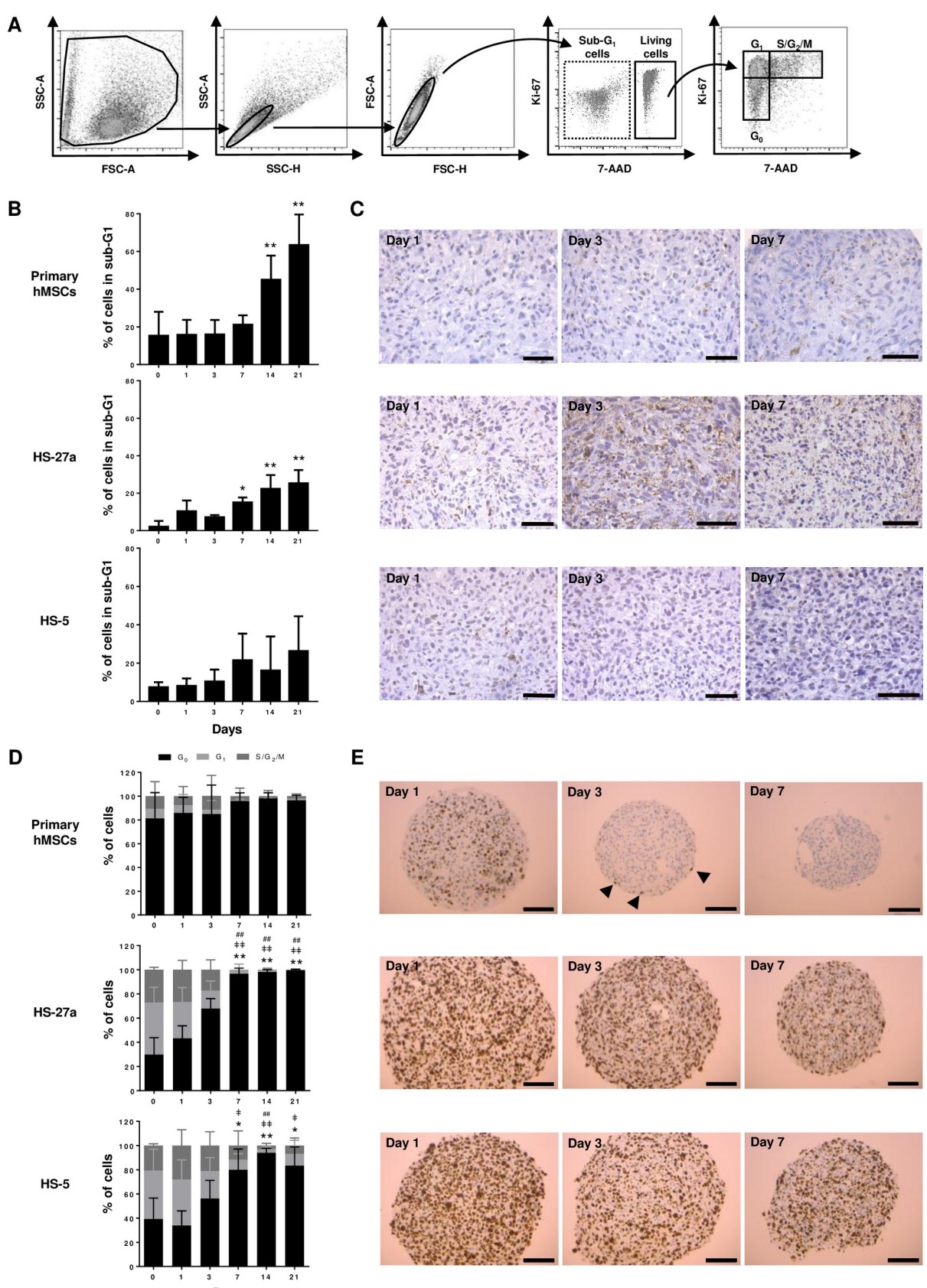

**Fig 4. Determination of proliferation and apoptosis of MSC-spheroids.** (A, B and D) Cell cycle analysis of spheroids over 21 days in culture. (A) Representative gating strategy from primary hMSCs at day 0, (B) sub-$G_1$ apoptosis quantification (primary hMSCs n = 6; HS-27a and HS-5 n = 3) and (D) cell cycle quantification (primary hMSCs n = 6; HS-27a and HS-5 n = 5; $^*$ for $G_0$; ‡ for $G_1$, # for S/$G_2$/M) (data are mean ± SD; $^*$, ‡, # compared to day 0; $^*$, ‡ p ≤ 0.05; $^{**}$, ‡‡, ## p ≤ 0.01). (C and E) Immunohistochemistry of (C) caspase-3 and (E) Ki-67 at days 1, 3 and 7 for primary hMSC-, HS-27a- and HS-5-spheroids (scale bars = 50 μm (C) and 100 μm (E)). Arrow heads indicate Ki-67-positive cells.

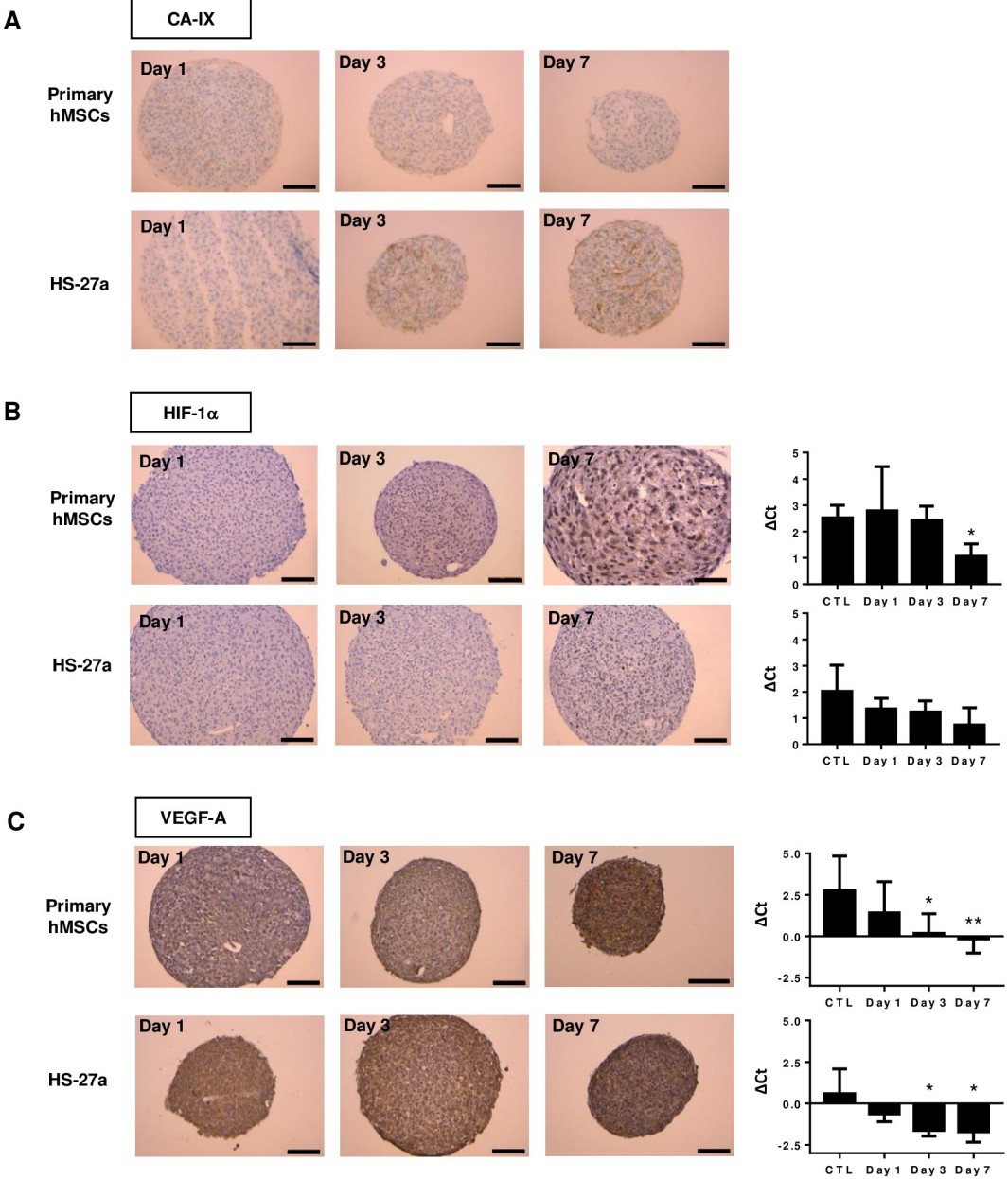

**Fig 5. Hypoxia detection of primary hMSC- and HS-27a-spheroids over 7 days in culture.** (A) Immunohistochemistry of CA-IX. (B) Immunohistochemistry and mRNA of HIF-1α. (C) Immunohistochemistry and mRNA expression of VEGF-A. (primary hMSCs n = 5; HS-27a n = 3; $^*$ p ≤ 0.05; $^{**}$ p ≤ 0.01; scale bars = 100 μm).

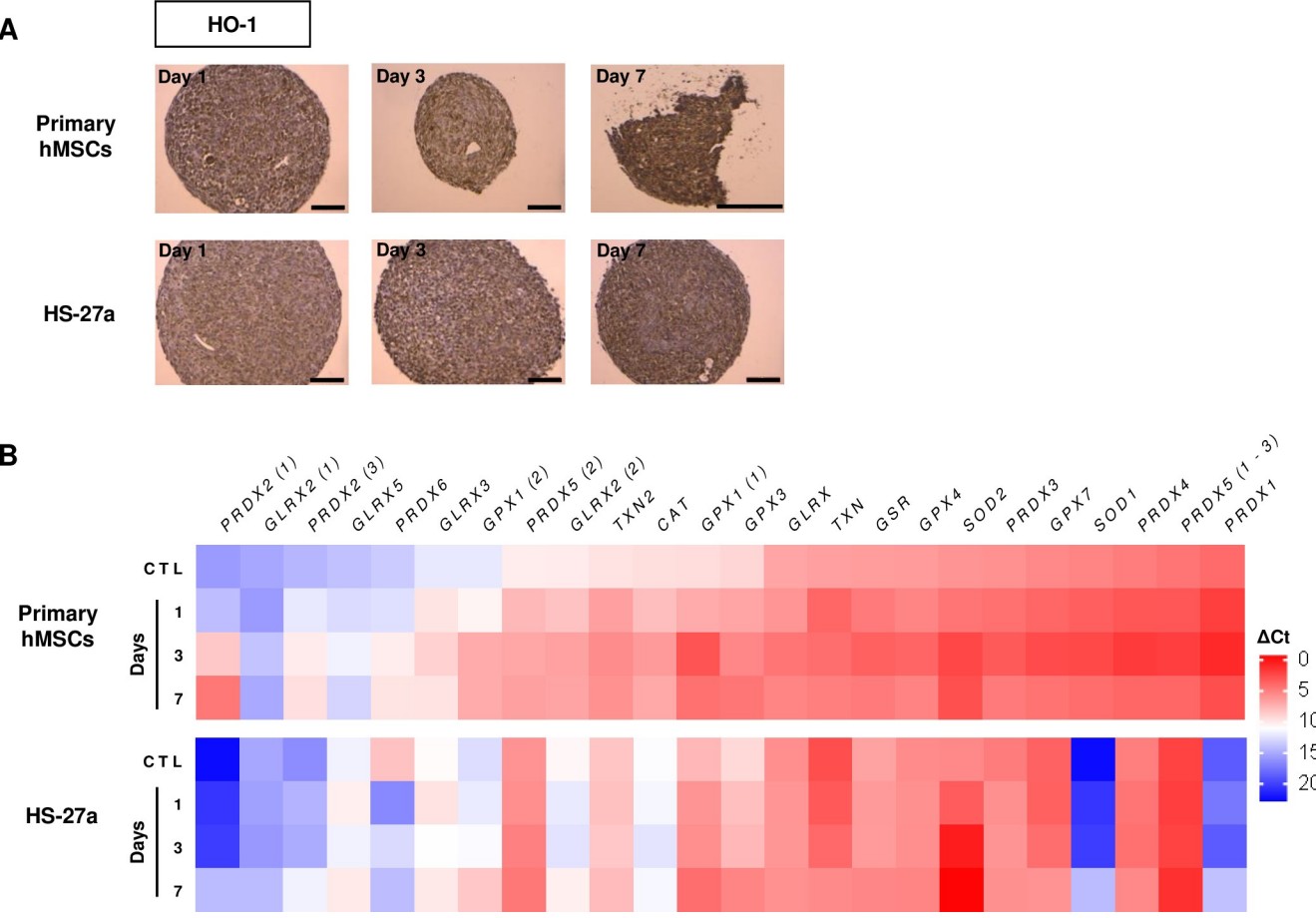

**Fig 6. Oxidative stress detection in primary hMSC- and HS-27a-spheroids.** (A) Immunohistochemistry of HO-1 (scale bars = 100 μm). (B) Expression of antioxidant genes (n = 3; data are mean; * compared to 2D control (CTL); * p ≤ 0.05; ** p ≤ 0.01).

oxygen species (ROS). Heme oxygenase 1 (HO-1) is induced by a variety of stressors, and is therefore a marker of hypoxia and oxidative stress [48,51]. Indeed, oxidative stress triggers nuclear relocation of NRF-2, a HO-1 transcription factor, which then leads to antioxidant response through induced expression of antioxidants by HO-1. In the spheroids, we observed a high expression of HO-1 at day 1, which increased over time (Fig 6A). Conversely, among the 24 antioxidant genes [52], we found a total of seven genes upregulated in spheroids from the primary hMSCs and the HS-27a cell line (Fig 6B). Remarkably, of these genes, four (*GPX1*, *PRDX2*, *SOD1* and *SOD2*) were commonly upregulated in both cell types irrespective of their initial expression level.

Together, these data indicate concomitant appearance of hypoxia and oxidative stress in both primary hMSC- and HS-27a-spheroids, which could therefore explain initial cell cycle arrest and further apoptosis in prolonged hypoxia.

## Dedifferentiation in MSC-spheroids

The 2D culture of MSCs critically leads to rapid loss of pluripotency after few passages, whereas MSC-spheroids can induce dedifferentiation, demonstrated by the expression of three pluripotent transcription factors (OCT-4, SOX-2 and NANOG) [32,53]. Furthermore, it has been described that hypoxia transcriptionally regulates these factors in a HIFs-dependent

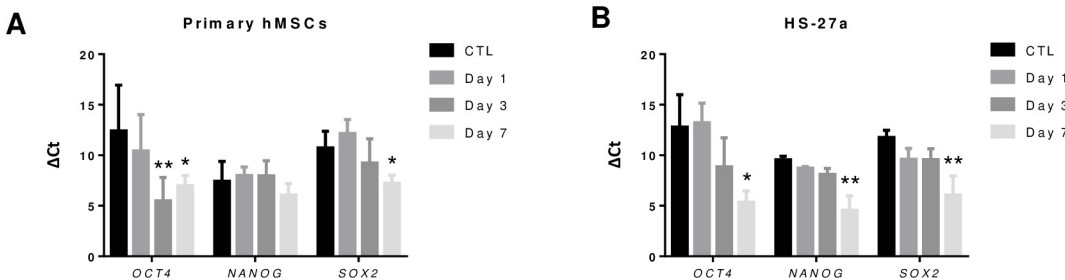

**Fig 7. Dedifferentiation detection of hMSC- and HS-27a-spheroids over 7 days in culture.** (A and B) Gene expression of *OCT4, NANOG* and *SOX2* for (A) primary hMSC- and (B) HS-27a-spheroids (primary hMSCs n = 5; HS-27a n = 3; *
p $\leq$ 0.05; ** p $\leq$ 0.01).

manner [54]. Therefore, in order to validate whether HS-27a behave similarly to primary hMSCs, we examined the expression of *OCT*-4, *SOX*-2 and *NANOG*, over time. Results showed that hMSC-spheroid formation was accompanied by upregulation of *OCT4* and *SOX2*, in agreement with previous studies, but surprisingly showed no upregulation of *NANOG* (Fig 7A). HS-27a had similar expression level of the three genes to hMSCs in 2D culture and had progressive increased expression of all markers, suggesting that the HS-27a retains dedifferentiation capacity like primary hMSCs (Fig 7B).

## Discussion

MSC-spheroids have mainly been designed to study biological processes such as those related to *ex vivo* stem cell maintenance, differentiation capacity, immunity or anti-inflammatory, with the aim to improve their use in regenerative medicine [25,55]. For instance, studies have demonstrated the benefit of using MSC-spheroids in treating cardiac, cerebral, kidney and hindlimb ischemia [56–59] or in tissue cartilage and bone repair [60–63]. MSCs are also highly involved in hematopoietic homeostasis [1,10]. However, until recently, they were viewed as rare heterogeneous populations lining vessels in the BM. Using multiscale 3D quantitative microscopy, Gomariz *et al.* have elegantly revealed that mesenchymal reticular subsets are remarkably more abundant than previously estimated [64]. This reinforces the interest of studying MSCs in the hematopoietic context and investigating them from a 3D angle.

Many studies have examined complex matrix-based 3D models, while ultimately, only few have approached MSC-spheroids as *in vitro* surrogate of the hematopoietic niche. In brief, HSCs can migrate and settle into MSC-spheroids that provide higher potential to promote HSCs expansion and stemness maintenance, compared to 2D co-cultures [18,19]. Direct intra-bone delivery of MSC-spheroids containing HSCs also sustained higher engraftment and retention of HSCs compared to injected HSCs [17]. However, Schmal *et al.* drew our attention on the fact that hanging drop method led to lower proliferation of hematopoietic cells and decreased generation of progenitors compared to 2D co-culture, although the stem cell capability has not been yet tested [35]. These contrary data suggest that spheroid colonization by HSCs may certainly depend on the method used to create spheroids. Spheroids are also attractive tools to uncover mechanisms by which malignant cells remodel their microenvironment and MSCs participate in chemoresistance and relapse [65]. As a proof of concept, Reagan *et al.* showed that MSC-spheroids help to unravel mechanisms of regulation of osteoblasts in multiple myeloma [66], while Aljitawi *et al.* demonstrated that chemotherapy is linked to the expression of N-cadherin in AML [23].

Although MSC-spheroids appear to be a promising tool, studies might have been limited by the availability of primary hMSCs and the reproducibility due to different sources. Strangely,

2D co-cultures with hematopoietic cells have long been established with cell lines, mostly murine, such as MS-5 or M2-10B4 [11], but they have almost never been used to create 3D aggregates. Our study provides an evaluation of the ability of three cell lines to form spheroids. We chose HS-27a and HS-5 cell lines for their human origin and their known capacity to sustain hematopoiesis [38]. The HS-5 cells are described as fibroblastoid cells, secreting high amount of growth factors that support hematopoietic progenitors proliferation. The HS-27a cell line has epithelioid morphology, producing low level of growth factors but supporting hemapoietic stem cells [38]. Unlike MS-5 cells, HS-5 and HS-27a cells do not retain contact inhibition that certainly, although of human origin, have limited their use for long-term cultures. Independently of the contact inhibition capacity, the three cell lines were able to provide quick and reproducible spheroids. The delay to achieve a complete spheroid, compared to primary hMSCs, could certainly be attributed to sedimentation speed, cell lines being much smaller than primaries that could hence sediment faster. In agreement with previous studies [18], we found that primary hMSCs provide highly cohesive spheroids with a smooth surface. MS-5 had a similar appearance, whereas spheroids from human cell lines were more disorganized with distinct cells at the surface. ECM deposition was visible for all types of spheroids but ECM composition could differ, and the higher proliferation of human cell lines, which certainly induces ECM remodeling, may explain the differences.

Contrary to primary hMSC-spheroids, all spheroids from cell lines kept a constant size over time. This could be partially explained for human cells by the fact that cell lines continue to proliferate in spheroid culture, whereas primary MSCs are mostly quiescent, and may compensate cell death. However, HS-27a showed a decrease in viable cells similarly to primary hMSCs and MS-5 are quiescent but do not shrink. Shrinking has already been reported for primary hMSCs [31,32,35,37,67–69] and has been attributed to autophagy [32]. Reduced cell growth or arrest is known to trigger autophagy [70]. Since, HS-27a and HS-5 cell lines proliferate until 7 days in spheroids, one could hypothesize that transformed cell lines may have lower autophagy, contrary to quiescent primary hMSCs. Consistent with this assumption, TEM revealed high amounts of cytoplasmic vacuoles and autophagosomes in primary hMSC-spheroids, as early as day 1. Although cytoplasmic vacuoles progressively appeared in spheroid from cell lines, almost no autophagosomes were detected. It is worth noting that HS-27a and HS-5 were obtained from primary hMSCs transformed with Human papillomavirus 16E6/E7, which activates autophagy via Atg9B and LAMP1 in cervical cancer cells [71]. However, neither TEM, nor LC3B staining showed autophagy in HS-27a cells. Indeed, LC3B expression was strongly induced in HS-27a cells, which indicates response to stress and induction of the autophagic process, but staining remains diffuse in the cytoplasm, suggesting a blockage of the autophagy process. However, diffuse LC3B staining may hamper the interpretation in IHC, while a dot-like staining patterns is more indicative of autophagy [41]. Dots may also reflect the accumulation of autophagosomes due to induction of autophagy, or due to inhibition of autophagy resulting from a lack of autophagosome degradation upon fusion with lysosomes [72]. In the absence of obvious autophagy in cell lines, cell death could therefore be explained by necrosis, as shown by SEM, and apoptosis observed by caspase-3 staining in IHC (early apoptosis) and 7-AAD staining in flow cytometry (late apoptosis and necrosis). Low apoptosis is detected for primary hMSC-spheroids, but massive lysed and necrotic cells are seen by SEM as early as day 1, in addition to autophagy. In agreement with our results, others studies have also demonstrated induction of apoptosis after several days [35,67,73]. While MS-5 cells resemble primary cells with low proliferation and strongly increased cell death, probably due to their contact inhibition, they conversely did not shrink like human cell lines, probably because of low or no autophagy.

Strong hypoxia and oxidative stress are among the stressors that could have induced autophagy and/or apoptosis in spheroids. Oxygen gradients have been frequently reported in tumor-spheres, with deep hypoxia surrounding a necrotic core [44–46], as well as in MSC-spheroids [47]. The hypoxia response mainly occurs through the stabilization of hypoxia-inducible factors (HIFs), which are regulators of multiple biological processes, such as angiogenesis or energetic metabolism. HIFs have an essential pro-survival role by promoting genes, such as those involved in metabolism and autophagy [50]. However, acute and prolonged hypoxia may also trigger cell death through blocking DNA replication and induced oxidative stress [44,46]. In MSC-spheroids, we found increased expression of hypoxia markers, including HIF-1α and its target CA-IX, VEGF-A, concomitant to induced oxidative stress, as revealed by increased expression of HO-1 and the antioxidant response. *SOD2* and *GPX1* were the two genes with the greatest upregulation, which indicates strong oxidative stress. Interestingly, although HO-1 and antioxidant genes are typically NRF-2 targets, they could also be regulated by HIFs. Altogether, these data indicate appearance of hypoxia and oxidative stress in primary hMSC- and HS-27a-spheroids, which could therefore explain cell cycle arrest, induction of autophagy and further apoptosis in prolonged hypoxia.

The 2D culture of MSCs critically leads to rapid loss of pluripotency after few passages, whereas 3D culture has showed greater MSC stemness maintenance, and induced dedifferentiation, demonstrated by the expression of three pluripotent transcription factors (OCT-4, SOX-2 and NANOG) and multipotent differentiation capacity [32,43,53,74]. It is worth noting that hypoxia favors stemness and dedifferentiation and induces expression of pluripotent transcription factors through HIFs [75,76]. Like primary hMSC-spheroids, HS-27a-spheroids had increased expression of pluripotent markers. This confirmed that HS-27a behave similarly to primary hMSCs and had preserved dedifferentiation capacity that could also be (re)activated during spheroid formation.

## Conclusions

Overall our data indicate that, like hMSCs, MSC cell lines can be used to make reproductible and easily handled spheroids. HS-27a cells resemble primary cells, and are of a particular interest for further studies, since they provide better support to HSCs compared to HS-5 cells [38–40]. Thus, this model could help in understanding mechanisms involved in MSC physiology and may be a simple model to study cell interactions in the hematopoietic niche. The model could also be extended to research metastatic process as previously described for breast cancer [28].

## Supporting information

**S1 Fig. Spheroids formation of mouse MS-5 cell line.** (A and B) Scanning electron microscopy (SEM) analysis over 14 days (scale bars = 100 μm (A) and 20 μm (B). (C) Sub-$G_1$ apoptosis quantification (n = 3) and (D) cell cycle quantification over 21 days in culture (n = 3; data are mean ± SD).
(TIF)

**S2 Fig. Transmission electron microscopy (TEM) observation of MSC-spheroids.** TEM analysis of primary hMSC-spheroids at day 1 (A), day 3 (B) and day 7 (C); Higher magnification is also shown to highlight autophagosomes. HS-27a-spheroids at day 1 (D), day 3 (E) and day 7 (F); HS-5-spheroids at day 1 (G), day 3 (H) and day 7 (I) and MS-5-spheroids at day 1 (J), day 3 (K) and day 7 (L). Scale bars = 20 μm.
(PPTX)

**S3 Fig. LC3B expression in HS-27a-spheroids.** Immunohistochemistry of LC3B is shown at days 1, 3 and 7 for HS-27a-spheroids (scale bars = 50 μm).
(TIF)

**S1 Video. A representative time-lapse video of spheroid formation.** 30 000 primary MSCs seeded into U-bottomed 96-well, in medium containing 0.5% of methylcellulose (Methocult™ SF H4236) were followed via a Nikon Eclipse TI-S microscope for 24 hours.
(MP4)

**S2 Video. A representative time-lapse video of spheroid formation.** 30 000 HS-27a cells seeded into U-bottomed 96-well, in medium containing 0.5% of methylcellulose (Methocult™ SF H4236) were followed via a Nikon Eclipse TI-S microscope for 24 hours.
(MP4)

**S3 Video. A representative time-lapse video of spheroid formation.** 30,000 HS-5 cells seeded into U-bottomed 96-well, in medium containing 0.5% of methylcellulose (Methocult™ SF H4236) were followed via a Nikon Eclipse TI-S microscope for 24 hours.
(MP4)

**S4 Video. A representative time-lapse video of spheroid formation.** 30,000 MS-5 cells seeded into U-bottomed 96-well, in medium containing 0.5% of methylcellulose (Methocult™ SF H4236) were followed via a Nikon Eclipse TI-S microscope for 24 hours.
(MP4)

**S1 Table. List of primers and probes sequences.**
(DOCX)

## Acknowledgments

We would like to thank P. G. Genever (University of York, UK) for providing valuable support, providing protocol and recommendations to establish spheroids. We also thank Sophie Hamard (University of Tours, France) for her technical assistance.

## Author Contributions

**Conceptualization:** Frédéric Mazurier.

**Data curation:** Margaux Deynoux, Nicola Sunter, Gaëlle Fromont, Frédéric Mazurier.

**Formal analysis:** Margaux Deynoux, Nicola Sunter, Hassan Dakik, Florence Rouleux-Bonnin, Philippe Roingeard, Gaëlle Fromont.

**Funding acquisition:** Olivier Hérault, Frédéric Mazurier.

**Investigation:** Margaux Deynoux, Nicola Sunter, Elfi Ducrocq, Hassan Dakik, Roseline Guibon, Julien Burlaud-Gaillard.

**Methodology:** Margaux Deynoux, Nicola Sunter, Elfi Ducrocq, Hassan Dakik, Roseline Guibon, Julien Burlaud-Gaillard, Lucie Brisson, Frédéric Mazurier.

**Project administration:** Frédéric Mazurier.

**Resources:** Louis-Romée le Nail, Olivier Hérault, Jorge Domenech, Gaëlle Fromont.

**Supervision:** Philippe Roingeard, Gaëlle Fromont, Frédéric Mazurier.

**Validation:** Margaux Deynoux, Nicola Sunter, Philippe Roingeard, Gaëlle Fromont, Frédéric Mazurier.

**Writing – original draft:** Margaux Deynoux, Nicola Sunter, Frédéric Mazurier.

**Writing – review & editing:** Margaux Deynoux, Nicola Sunter, Hassan Dakik, Roseline Guibon, Julien Burlaud-Gaillard, Lucie Brisson, Florence Rouleux-Bonnin, Olivier Hérault, Jorge Domenech, Philippe Roingeard, Gaëlle Fromont, Frédéric Mazurier.

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
