## [Decision Letter · Decision Letter 0]

14 Jan 2020

PONE-D-19-30255

A comparative study of the capacity of mesenchymal stromal cell lines to form spheroids

PLOS ONE

Dear Dr Mazurier,

Thank you for submitting your manuscript to PLOS ONE. After careful consideration, we feel that it has merit but does not fully meet PLOS ONE’s publication criteria as it currently stands. Therefore, we invite you to submit a revised version of the manuscript that addresses the points raised during the review process.

We would appreciate receiving your revised manuscript by Feb 28 2020 11:59PM. To enhance the reproducibility of your results, we recommend that if applicable you deposit your laboratory protocols in protocols.io, where a protocol can be assigned its own identifier (DOI) such that it can be cited independently in the future. For instructions see: http://journals.plos.org/plosone/s/submission-guidelines#loc-laboratory-protocols

We look forward to receiving your revised manuscript.

Kind regards,

Atsushi Asakura, Ph.D

Academic Editor

PLOS ONE

Journal Requirements:

"Primary normal BM-MSCs were isolated from healthy donors (without any hematological disorder) undergoing orthopedic surgery (University Hospital, Tours,

France) after informed consent and following a procedure approved by the local ethical committee."

in the Methods section of the manuscript, please add the same text to the “Ethics Statement” field of the submission form (via “Edit Submission”).

Additional Editor Comments (if provided):

Reviewers' comments:

Reviewer's Responses to Questions

**Comments to the Author**

1. Is the manuscript technically sound, and do the data support the conclusions?

Reviewer #1: Yes

Reviewer #2: Partly

Reviewer #3: Partly

2. Has the statistical analysis been performed appropriately and rigorously? 

Reviewer #1: I Don't Know

Reviewer #2: Yes

Reviewer #3: Yes

3. Have the authors made all data underlying the findings in their manuscript fully available?

Reviewer #1: No

Reviewer #2: Yes

Reviewer #3: Yes

4. Is the manuscript presented in an intelligible fashion and written in standard English?

Reviewer #1: Yes

Reviewer #2: Yes

Reviewer #3: Yes

5. Review Comments to the Author

Reviewer #1: In this manuscript, the authors used spheroids to study differences of primary human mesenchymal stem cells (hMSCs) and immortalized mesenchymal stem cell (MSC) lines. Overall, it seems that the study properly designed to achieve their research goal. However, the current manuscript still has a number of major and minor concerns and needs better clarifications in several places.

Major concerns:

1. The rationale behind the study is a bit unclear. The authors should describe more how MSC-based spheroids are specifically available to study the hematopoietic niche. The sentences in the first paragraph of the Discussion section are vague or not specific for MSC-based spheroids.

2. It is expected that the cells in each well gradually formed a single spheroid at the end. However, in Figures1B, 2A, and Supplementary Videos, small cell aggregates are identified near a large spheroid. Please clarify how these aggregates might influence the analysis in this study. For instance, were these small aggregates included (or excluded) when the number of cells was analyzed in Figure 2C?

3. Figure 2B: As this is 3D culture, the spheroid volume may sound more reasonable compared to the perimeter.

4. Figure 3: Additional indications or labeling should be used to point out specific histological features. The figure legend should also be expanded more, not only describing the results in the text. In Figure 3C, where is the appearance of a progressive cell injury?. By the way, what does “a progressive cell injury” mean?

5. Figure 4: Like Ki67 staining, it is worth to have additional immunohistochemical results showing cell death in the spheroids.

6. Figure 6B: Although the results are summarized as a heat map without specific values, statistical differences are presented there. Is this appropriate?

7. Page 16, the section named “Stemness in MSCs-derived spheroids”: The word “stemness” sounds too definitive here, as the conclusion was only supported by the results of gene expression for specific stem cell markers.

8. The Discussion section may be short a bit, when compared to the amount of experiments and results.

9. The abbreviations “MSCs-“ or “MSCs-derived” are used in many places. They should be “MSC-“ and “MSC-derived”.

Minor concerns:

1. Page 3, Line 54 from the bottom: Please spell out “2D” as this abbreviation comes at the first place in the main text.

2. An abbreviation “3D” should be defined with “three-dimensional” in Page 3, Line 55.

3. Page 5, Line 90: Please describe the dose of FGF-2 with nanogram or microgram/mL. A use of % is not common and not helpful for readers.

4. Page 5, Line 93: The passage number of established MSC lines (“between passage 5 and 20”) would not be accurate, as these cells were already expanded in culture before the authors had received them from the repositories.

5. Page 8, Line 151-158, Quantitative real-time PCR: Why does only this section have the catalogue numbers for the products?

6. Page 9, Line 170-173, Statistical analysis: Nonparametric statistics are commonly used. I understand that in general nonparametric methods are not easy to achieve a statistical difference when the number of subjects is relatively small like n=3 or 4. This may be still acceptable, but please justify why these statistical methods were specifically applied.

7. Page 9, Line 180-181: Please add “(MethoCult H4100 and SF H4236)” after “two commercial methylcelluloses.” How are these two products different in terms of components?

8. Page 14, Line 281, Figure 4D legend: Please revise “Arrows” to “Arrow heads.”

9. Page 15, Line 309: What does “(Patent WO2016083742)” mean?

10. Page 15, Line 320: The word “established MSCs” is vague. Be specific.

11. Page 16, the word “stemness”: What retains stemness is the cells cultured in spheroids, not spheroids themselves.

12. These words should be reconsidered in use: 3D MSC structures (Page 4, Line 60), gold-standard (Page 4, Line 74), a stemness capacity (Page 16, Line 334-335), stemness detection (Page 16, Line 337).

Reviewer #2: In this study, the authors examined the sphere-forming capacity of two human bone marrow stromal cell lines, morphologically resembling epithelial cell or fibroblast, and a murine bone marrow stromal cell line, comparing to primary human bone marrow-derived MSCs. An approach based on cell aggregation in methylcellulose-based medium was used. The size of the sphere, the number of cells constituting the sphere, morphology, cell cycle, and gene expression related to hypoxia-induced stress response were examined chronologically. Although it is interesting to know them, honestly, it is not clear to me what model the authors tried to create. A simple model to evaluate sphere-forming capacity of cells or searching cell line exhibiting MSC-like cellularity in terms of sphere formation capacity? It has been reported that in comparison with dissociated cells or cells expanded on adhesion culture, MSC spheroids exhibit improved survival and secretion of trophic factors while maintaining or enhancing their differentiation capacity. The reviewer somehow thinks that there is something missing.

Major concern

If the authors are trying to find a cell line exhibiting MSC-like cellularity in terms of sphere formation capacity, a single cell culture may be needed to assess sphere forming capacity. Proof of MSC-like cellularity such as self-renewability, tri-linage differentiation capacity is, of course, necessary.

Reviewer #3: Although the manuscript is interesting in the comparison of the capacity in the spheroid culture from three types of hMSCs there are some serious problems. The authors should be addressing them.

Special comments

1) It totally is the serious problem that the authors did not examine the difference in the differentiation ability into osteoblasts, adipocytes, and chondrocytes from spheroid with primary or cell line hMSCs in the present experiments because it is an important role of regeneration ability for cellular therapy using hMSCs.

2) Materials and Methods: There is no ethics statement (permission number) and preparation and condition of primary hMSCs. Did the authors conducted in compliance with Declaration of Helsinki and get the informed consent from patients? Furthermore, did the authors separate and collect the primary hMSCs with stemness makers (CD29, CD44, and CD105 etc.) or not (heterogeneity) in present experiments?

3) Results: Data interpretation involved in Results section. The authors should describe them in Discussion section.

4) Results: The authors stated the primary hMSCs was less optimal for spheroid culture. It is well known that the proliferation and maintenance of spheroid culture is dependent on the number of spread cells. Did the authors should examine and confirm the less number of spread cells or diameter less than 300 micrometer without shrinking spheroid in primary hMSCs? Furthermore, the authors should examine whether the smaller spheroid derived from primary hMSCs induced activation of HIF-1alpha and apoptosis or not.

5) Figure1: Why did the authors use the two types of methylcelluloses? What is the difference, such as components, viscosity, and moisturization etc.? The authors should explain them.

6) Figure 2: It confuses the murine or human hMSCs. In Fgure2, the authors should add the murine MS-5 data with perimeter and cell number/spheroid, and culture days, but not its supplemental video.

7) Discussion: The authors explain the induction of autophagy in primary hMSCs. However, the authors did not check it. The authors should check the expression of autophagy related molecules, such as LCII etc.

Minor comments

8) Results p10, line 204: It confuses the primary or immortalization hMSCs. The authors should add the word primary MSCs.

9) Discussion p17, line 352: Was it inserted the reference, not referenced number?

10) Fig3C and Fig.5 were low density images, then it is difficult to interpret them.

6. PLOS authors have the option to publish the peer review history of their article (what does this mean?). If published, this will include your full peer review and any attached files.

Reviewer #1: No

Reviewer #2: No

Reviewer #3: No

---

## [Author Response · Author response to Decision Letter 0]

16 Apr 2020

Rebuttal letter [PONE-D-19-30255]

April 6th, 2020

Dear Editor,

First, we would like to thank the editor and reviewers for their remarks and hope that we have now answered most of their concerns and improved the manuscript to fully meet PLOS ONE’s publication criteria. To make it easier, our answers are in blue.

Journal Requirements:

PLOS ONE’s requirements have been carefully rechecked, and errors are corrected.

"Primary normal BM-MSCs were isolated from healthy donors (without any hematological disorder) undergoing orthopedic surgery (University Hospital, Tours, France) after informed consent and following a procedure approved by the local ethical committee."

In the Methods section of the manuscript, please add the same text to the “Ethics Statement” field of the submission form (via “Edit Submission”).

We apologize for the lack of information and now provide a more detailed sentence (page 5, line 82): “Primary BM hMSCs were obtained by iliac crest aspiration from healthy donors (without hematological disorders) undergoing orthopedic surgery at the University Hospital of Tours, after informed consent, for cell banking according to the Declaration of Helsinki, as approved by the French Ministry of Education and Research (authorization number No. DC-2008-308)”. Please note that the local ethical committee does not provide permission number.

The phrase was « Both techniques worked well for primary hMSCs but the second was more appropriated for further analyses and offered lesser dehydration (data not shown) ». This was a simple observation that we though informative for readers. We understand the general requirement and removed the statement. The sentence (page 11, line 193) is now « Both techniques worked well for primary hMSCs but the second was more appropriate for further analyses since the handling is easier and the volume of medium higher, which could prevent starvation and dehydration in long-term cultures ».

Review Comments to the Author

Reviewer #1: In this manuscript, the authors used spheroids to study differences of primary human mesenchymal stem cells (hMSCs) and immortalized mesenchymal stem cell (MSC) lines. Overall, it seems that the study properly designed to achieve their research goal. However, the current manuscript still has a number of major and minor concerns and needs better clarifications in several places.

Major concerns:

1. The rationale behind the study is a bit unclear. The authors should describe more how MSC-based spheroids are specifically available to study the hematopoietic niche. The sentences in the first paragraph of the Discussion section are vague or not specific for MSC-based spheroids.

We understand the criticism of the reviewer and the need of clarification. Our work is not limited to hematopoietic niche research, although we are convinced about the real need, but more widely of interest for any kind of specific issue including those in regenerative medicine that require spheroids.

Since the Reviewer had also found the discussion to short (concern #8), we decided to fully reformat the discussion and develop the use of MSC-derived spheroids in hematological research (Page 20, Discussion). We believe that would answer both criticisms.

2. It is expected that the cells in each well gradually formed a single spheroid at the end. However, in Figures1B, 2A, and Supplementary Videos, small cell aggregates are identified near a large spheroid. Please clarify how these aggregates might influence the analysis in this study. For instance, were these small aggregates included (or excluded) when the number of cells was analyzed in Figure 2C?

In Figure 1B, SF H4236 with 0.5 and 1 % provided obviously a unique sphere. This is why for further experiments only the SF H4236 condition was retained. This was mentioned in the text (page 11, line 195) “The SF H4236 methylcellulose at a concentration of 0.5 % was adopted because it generated one spheroid per well with lower condensation aspect for primary hMSCs (Fig 1B)”. In this figure, cells were not enumerated.

In figure 2A, very few aggregates are seen for MS-5 and a bit of spreading for HS-5 only after 14 days. Small aggregates occurred from time to time, certainly due to remaining debris (medium, plastic), but it was very marginal. That appears, however, when we acquired videos. Although the low number of cells in tiny aggregates may account for a low percentage and in all conditions, we decided at the beginning to take into consideration only wells with unique spheroids and no spreading. Therefore, the count in Figure 2C is from fully formed spheroid. We do understand the concern of the reviewer and thus modified our sentence in Materials and Methods section (page 6, line 107) “To determine the number of cells in each spheroid over time, only wells with unique, fully-formed spheroid were selected ”.

3. Figure 2B: As this is 3D culture, the spheroid volume may sound more reasonable compared to the perimeter.

Since spheroids are not perfectly rounded and may sometime be slightly ovoid (see figures 2A and 3A), we believed that giving the perimeters, even as arbitrary unit values, will be more exact and rigorous than any other parameter. Volume will require diameter measurement and be a rough estimation by applying the same formula “4/3 x π x R3” on all spheroids considering they are round, otherwise we will have to measured length and width, with unknown height. Thus, the difference between spheres will be almost the same at the end but with increased technical and calculation errors. So, we think that this is useless and certainly not reasonable to do. In the literature, volume is not frequently provided, whereas authors mostly show diameter (Pennock et al., Schmal et al., Tsai et al., De Barros et al., etc.), which may introduce a bias as abovementioned.

4. Figure 3: Additional indications or labeling should be used to point out specific histological features. The figure legend should also be expanded more, not only describing the results in the text. In Figure 3C, where is the appearance of a progressive cell injury? By the way, what does “a progressive cell injury” mean?

We agree with the reviewer. We kept only SEM as main figure, pointed out specific histological features and consequently modified the legend. To get a better resolution, we had to transfer TEM in supplemental material, as S2 Fig, with individual photos for each spheroid type from day 1 to day 7. “Progressive cell injury” is illustrated by loss of cell-cell adhesion, swelling of the cytoplasm, increased lysed cells and cell death. This has been changed in the text/legend due to figure modifications (page 14, line 253).

5. Figure 4: Like Ki67 staining, it is worth to have additional immunohistochemical results showing cell death in the spheroids.

We initially though that death shown by electronic microscopy and flow cytometry would be enough. We, however, agree with the reviewer that it is worth showing IHC. We therefore included caspase 3 staining, which could detect much earlier induction of apoptosis than 7-AAD staining. Results were added as figure 4C and reported in the text (page 15, line 281).

6. Figure 6B: Although the results are summarized as a heat map without specific values, statistical differences are presented there. Is this appropriate?

To summarize the high amount of transcriptomic data, we believe that heatmap might be better representation than multiple histograms, but thus lost the possibility of showing statistics. Possibly not classically, we believed we had found a way that might be the most informative to the reader and rigorously important to prove which of the gene is statistically different by adding a star. So, as it does not seem appropriate, we do not mind removing stars.

7. Page 16, the section named “Stemness in MSCs-derived spheroids”: The word “stemness” sounds too definitive here, as the conclusion was only supported by the results of gene expression for specific stem cell markers.

We agree with the reviewer. The reviewer 3 also mentioned that in his concern #11/12. We, therefore, modified the paragraph (page 18, line 345), excluding the word stemness and dedifferentiation, which seems more suitable.

8. The Discussion section may be short a bit, when compared to the amount of experiments and results.

We agree. We reformatted and increased the Discussion section (See also concern #1).

9. The abbreviations “MSCs-“ or “MSCs-derived” are used in many places. They should be “MSC-“ and “MSC-derived”.

We thank the reviewer for drawing our attention to this error. This is now corrected.

Minor concerns:

1. Page 3, Line 54 from the bottom: Please spell out “2D” as this abbreviation comes at the first place in the main text.

We thank the reviewer for drawing our attention on this error. It is now modified (page 3, line 54).

2. An abbreviation “3D” should be defined with “three-dimensional” in Page 3, Line 55.

It has been also modified (page 3, line 55).

3. Page 5, Line 90: Please describe the dose of FGF-2 with nanogram or microgram/mL. A use of % is not common and not helpful for readers.

This is true and has been changed for “….1 ng/mL of recombinant human FGF basic” (page 5, line 90).

4. Page 5, Line 93: The passage number of established MSC lines (“between passage 5 and 20”) would not be accurate, as these cells were already expanded in culture before the authors had received them from the repositories.

The reviewer is right. We thus removed this description and only kept the information for primary “homemade” cells (page 5, line 92).

5. Page 8, Line 151-158, Quantitative real-time PCR: Why does only this section have the catalogue numbers for the products?

This was an error. We removed numbers (page 9, line 159).

6. Page 9, Line 170-173, Statistical analysis: Nonparametric statistics are commonly used. I understand that in general nonparametric methods are not easy to achieve a statistical difference when the number of subjects is relatively small like n=3 or 4. This may be still acceptable, but please justify why these statistical methods were specifically applied.

Our sampling was always lower than n = 6, which does not allow us to make parametric statistics. Therefore, we used nonparametric tests. We added this notion in the ‘Materials and methods” (page 10, line 178).

7. Page 9, Line 180-181: Please add “(MethoCult H4100 and SF H4236)” after “two commercial methylcelluloses.” How are these two products different in terms of components?

All information is available online but to clarify we added a sentence in the “Materials and Methods” section: “Both medium contains methylcellulose in IMDM, but SF H4236 is supplemented with bovine serum albumin, recombinant human insulin, human transferrin (iron-saturated), 2-Mercaptoethanol and unknown supplements as described by the manufacturer”. (page 6, line 101).

8. Page 14, Line 281, Figure 4D legend: Please revise “Arrows” to “Arrow heads.”

We agree. This has been changed (page 16, line 304).

9. Page 15, Line 309: What does “(Patent WO2016083742)” mean?

The list of genes was included in a patent from our laboratory, it is now published in a paper as indicated by the following reference [52]: “Picou et al., Bone marrow oxidative stress and specific antioxidant signatures in myelodysplastic syndromes. Blood Adv. 2019 Dec 23;3(24):4271-4279. doi: 10.1182/bloodadvances.2019000677”. This is now cited changed (page 18 line 333).

10. Page 15, Line 320: The word “established MSCs” is vague. Be specific.

We agree. The words “established MSCs” has been changed to “both primary hMSC- and HS-27a-spheroids” (page 18 line 342).

11. Page 16, the word “stemness”: What retains stemness is the cells cultured in spheroids, not spheroids themselves.

We agree and made modifications in the title and in the text of this paragraph. We wrote dedifferentiation instead of stemness (See major concern #7) (pages 18, line 345 and following).

12. These words should be reconsidered in use: 3D MSC structures (Page 4, Line 60), gold-standard (Page 4, Line 74), a stemness capacity (Page 16, Line 334-335), stemness detection (Page 16, Line 337).

The terms condensates or aggregates are indeed more often used in the literature. We therefore changed (page 4, line 59). We are not sure about what the reviewer means by reconsidering gold-standard but changed it for “the currently used murine MS-5 cell line” instead (page 4, line 73). As mentioned above, the text including stemness was replaced by dedifferentiation.

Reviewer #2: In this study, the authors examined the sphere-forming capacity of two human bone marrow stromal cell lines, morphologically resembling epithelial cell or fibroblast, and a murine bone marrow stromal cell line, comparing to primary human bone marrow-derived MSCs. An approach based on cell aggregation in methylcellulose-based medium was used. The size of the sphere, the number of cells constituting the sphere, morphology, cell cycle, and gene expression related to hypoxia-induced stress response were examined chronologically. Although it is interesting to know them, honestly, it is not clear to me what model the authors tried to create. A simple model to evaluate sphere-forming capacity of cells or searching cell line exhibiting MSC-like cellularity in terms of sphere formation capacity? It has been reported that in comparison with dissociated cells or cells expanded on adhesion culture, MSC spheroids exhibit improved survival and secretion of trophic factors while maintaining or enhancing their differentiation capacity. The reviewer somehow thinks that there is something missing.

Major concern

If the authors are trying to find a cell line exhibiting MSC-like cellularity in terms of sphere formation capacity, a single cell culture may be needed to assess sphere forming capacity. Proof of MSC-like cellularity such as self-renewability, tri-linage differentiation capacity is, of course, necessary.

It is not clear for us what the reviewer means by “MSC-like cellularity in terms of sphere formation capacity?”. Our purpose never aimed to identify/compare stem cell or differentiation capacities of cell lines, but rather to determine whether, like primary hMSCs, they can form spheroids. HS-27a and HS-5 human are transformed cells. What will be the purpose of testing for their self-renewability ability? Both are capable of tri-lineage differentiation in 2D (1. Ischac et al., submitted manuscript Figure 1; 2. Liu et al., 2015; 3. Vallet et al., 2011). We could certainly expect them to do the same in 3D as figure 2, but in modified medium.

[For figures please check Cover Letter file]

1. Figure 1. Differentiation potential of HS-27a and HS-5 cells into osteoblasts, adipocytes, chondrocytes and vascular smooth muscle. Scale bars indicate 100 µm. Images from Ischac et al., submitted manuscript.

2. Liu, B., Wu, S., Han, L., Zhang, C., 2015. Β-Catenin Signaling Induces the Osteoblastogenic Differentiation of Human Pre-Osteoblastic and Bone Marrow Stromal Cells Mainly Through the Upregulation of Osterix Expression. Int. J. Mol. Med. 36, 1572–82. doi:10.3892/ijmm.2015.2382

3. Vallet, S., Pozzi, S., Patel, K., Vaghela, N., Fulciniti, M.T., Veiby, P., Hideshima, T., Santo, L., Cirstea, D., Scadden, D.T., Anderson, K.C., Raje, N., 2011. A Novel Role for CCL3 (MIP-1α) in Myeloma-induced Bone Disease via Osteocalcin Downregulation and Inhibition of Osteoblast Function. Leukemia 25, 1174–81. doi:10.1038/leu.2011.43.

MSC-spheroids are mainly developed in a context of bioengineering, to study impact on survival, differentiation capacity, etc. They were also suggested as in vitro surrogate models for the hematopoietic bone marrow microenvironment. Therefore, using cell lines may help in investigating mechanisms in a simple system whatever is the research domain, bioengineering or hematopoiesis, in the absence of primary hMSCs. Nothing has been previously published on cell lines in 3D, whereas cells lines are often used in 2D to support hematopoiesis, for instance. To our knowledge, all studies were based on MSC-aggregates, using different matrix and methods, and not on single cells. Probably few data such as those from Isern et al. used human single cells in order to study their stem cell capacity, and others on murine cells, possibly. We fit our work on methods reported before.

The authors are very sorry and apologize if the reviewer did not pick up the message and hope that the manuscript is more comprehensible now.

Reviewer #3: Although the manuscript is interesting in the comparison of the capacity in the spheroid culture from three types of hMSCs there are some serious problems. The authors should be addressing them.

Special comments

1) It totally is the serious problem that the authors did not examine the difference in the differentiation ability into osteoblasts, adipocytes, and chondrocytes from spheroid with primary or cell line hMSCs in the present experiments because it is an important role of regeneration ability for cellular therapy using hMSCs.

As mentioned above, HS27a and HS5 are known to be able to differentiate into the tri-lineages. We agree that might be worth to test in 3D, but this would require specific media, different than the one we used. In addition, we wondered at which timepoint the reviewer think it will be relevant to perform differentiation. Alternately, based on protocol from the literature, direct differentiation and staining could be performed directly on spheroids (Ref 74 - Cheng et al. 2012), by adding specific growth factors in our medium in order to obtain osteoblasts, chondrocytes and adipocyte differentiation. Before closing our laboratory, due to Covid-19, we were only able to show adipocytic differentiation of the HS-27a cell line (see figure 2). However, we thought that would, first, change the purpose of our main message of our actual manuscript and, second, require to set up much more methods, particularly to get osteogenic and chondrogenic differentiation as for 2D (Figure 1, reviewer#2 concerns). Thus, this would certainly be of interest for a further story.

[For figures please check Cover Letter file]

Figure 2. Adipogenic differentiation of HS-27A cells revealed after 2 weeks by neutral lipid vacuoles stained with Nile Red (Yellow).

2) Materials and Methods: There is no ethics statement (permission number) and preparation and condition of primary hMSCs. Did the authors conducted in compliance with Declaration of Helsinki and get the informed consent from patients?

We agree with the reviewer, the original sentence already mentioned “informed consent” but was not complete. This has been corrected. Please refer to above Journal requirements, concern #2.

Furthermore, did the authors separate and collect the primary hMSCs with stemness makers (CD29, CD44, and CD105 etc.) or not (heterogeneity) in present experiments?

MSCs from BM aspirates were seeded and cultured in current culture conditions with no additional cell staining and sorting. The protocol has been expanded in “material and methods” (page 5, line 86): “Samples from BM aspirates were diluted in MEM Alpha (Life Technologies, Villebon-sur-Yvette, France) and filtered through a cell-strainer prior centrifugation (350 x g, 10 min). Cells were resuspended and seeded at 105 to 2.105 cells/cm2 in MEM Alpha supplemented with 100 U/mL penicillin and 100 µg/mL streptomycin (both from Life Technologies) and 1 ng/mL of recombinant human FGF basic (FGF-2, R&D Systems, Abingdon, United Kingdom). Medium was changed twice a week until cells reached confluency”. 

3) Results: Data interpretation involved in Results section. The authors should describe them in Discussion section.

We do not understand the concern. Could you please precise which data interpretation? Should we remove some from results or discuss in discussion section? We did not find much interpretation in results and describe mostly everything in the discussion. The discussion has been, however, rewritten to answer the concerns of reviewer 1.

4) Results: The authors stated the primary hMSCs was less optimal for spheroid culture.

It is not clear for us what the reviewer means by that. Less optimal than what? Cell lines? If so, it is not what we wished to state, or we were misunderstood. We seek for a cell line that should replace, somehow, the primary hMSCs with similar properties.

It is well known that the proliferation and maintenance of spheroid culture is dependent on the number of spread cells. Did the authors should examine and confirm the less number of spread cells or diameter less than 300 micrometer without shrinking spheroid in primary hMSCs? Furthermore, the authors should examine whether the smaller spheroid derived from primary hMSCs induced activation of HIF-1alpha and apoptosis or not.

We do not see spreading, except from time to time with HS-5 after 14 days, and were not interested in this aspect. Since this phenomenon is not observed, it does not seem relevant to look at smaller spheroids. Our method is based on a defined number of cells, which provided spheroids with an average diameter ≥ 300 �m, why should we start with lower number of cells to get smaller spheroids? We might not have completely understood the real meanings of this remark.

5) Figure1: Why did the authors use the two types of methylcelluloses? What is the difference, such as components, viscosity, and moisturization etc.? The authors should explain them.

We were looking for commercial methylcelluloses without cytokines that could be further used for hematopoietic cells. StemCell Technologies provides 3 products without cytokines: SF H4236, H4100 and H4230. After checking their composition, and based on other works, we excluded the H4230 because it contains fetal bovine serum (FBS), and we wished to use our own FBS selected for optimal growth of MSCs. SF H4236 has the same base composition as H4100, but also contains BSA, Insulin, transferrin and other unknown components. A sentence has been added in the Material and Methods (page 6, line 101): “Both media contains methylcellulose in IMDM, but SF H4236 is supplemented with bovine serum albumin, recombinant human insulin, human transferrin (iron-saturated), 2-Mercaptoethanol and unknown supplements as described by the manufacturer ”.

6) Figure 2: It confuses the murine or human hMSCs. In Figure2, the authors should add the murine MS-5 data with perimeter and cell number/spheroid, and culture days, but not its supplemental video.

Following the reviewer recommendations, this has been changed. Figure 2 now includes MS-5 data.

7) Discussion: The authors explain the induction of autophagy in primary hMSCs. However, the authors did not check it. The authors should check the expression of autophagy related molecules, such as LCII etc.

We agree with the reviewer and modified figures and text. As mentioned above (reviewer 1, concern #4) TEM images with higher magnification are chronologically shown in S2 fig. This figure shows autophagosome formation only in primary hMSC-spheroids, as soon as day1. We also stained HS-27a-spheroids for LC3B, as recommended by the reviewer, and found, although triggered, only diffuse cytoplasmic expression that do not reflect autophagosome formation as shown in S3 fig. LC3B detected by IHC, still seems under debate. This was underlined now in the “Electronic microscopy” chapter as well as in the discussion (page 22, line 422): “…diffuse LC3B staining may hamper the interpretation in IHC, while a dot-like staining patterns is more indicative of autophagy [41]. Dots may also reflect the accumulation of autophagosomes due to induction of autophagy, or due to inhibition of autophagy resulting from a lack of autophagosome degradation upon fusion with lysosomes [72]”. 

Minor comments

8) Results p10, line 204: It confuses the primary or immortalization hMSCs. The authors should add the word primary MSCs.

We agree with the reviewer and changed the sentence for (page 12, line 207): “The spheroid-forming capacity was followed for two human cell lines, HS-27a and HS-5, and compared to that of primary hMSCs. The two cell lines have been obtained by immortalization of hMSCs from the same BM sample with the papilloma virus E6/E7 genes”.

9) Discussion p17, line 352: Was it inserted the reference, not referenced number?

We thank the reviewer for drawing our attention on this error. This has been changed.

10) Fig3C and Fig.5 were low density images, then it is difficult to interpret them.

We understand but the original has better quality. Due to the limited size required by the journal, we had to make it as low density. We now move fig 3C to supplemental data with its original size and intend to keep a better quality for fig 5.

---

## [Decision Letter · Decision Letter 1]

18 May 2020

A comparative study of the capacity of mesenchymal stromal cell lines to form spheroids

PONE-D-19-30255R1

Dear Dr. Mazurier,

We are pleased to inform you that your manuscript has been judged scientifically suitable for publication and will be formally accepted for publication once it complies with all outstanding technical requirements.

With kind regards,

Atsushi Asakura, Ph.D

Academic Editor

PLOS ONE

Additional Editor Comments (optional):

Reviewers' comments:

Reviewer's Responses to Questions

**Comments to the Author**

1. If the authors have adequately addressed your comments raised in a previous round of review and you feel that this manuscript is now acceptable for publication, you may indicate that here to bypass the “Comments to the Author” section, enter your conflict of interest statement in the “Confidential to Editor” section, and submit your "Accept" recommendation.

Reviewer #1: All comments have been addressed

Reviewer #3: All comments have been addressed

2. Is the manuscript technically sound, and do the data support the conclusions?

Reviewer #1: Yes

Reviewer #3: Yes

3. Has the statistical analysis been performed appropriately and rigorously? 

Reviewer #1: Yes

Reviewer #3: Yes

4. Have the authors made all data underlying the findings in their manuscript fully available?

Reviewer #1: Yes

Reviewer #3: Yes

5. Is the manuscript presented in an intelligible fashion and written in standard English?

Reviewer #1: Yes

Reviewer #3: Yes

6. Review Comments to the Author

Reviewer #1: (No Response)

Reviewer #3: The manuscript almost had revised in the pointed-out parts. I recommend the paper to publish for your journal.

7. PLOS authors have the option to publish the peer review history of their article (what does this mean?). If published, this will include your full peer review and any attached files.

Reviewer #1: No

Reviewer #3: No

---

## [Editor Report · Acceptance letter]

22 May 2020

PONE-D-19-30255R1 

A comparative study of the capacity of mesenchymal stromal cell lines to form spheroids 

Dear Dr. Mazurier:

I am pleased to inform you that your manuscript has been deemed suitable for publication in PLOS ONE. Congratulations! Your manuscript is now with our production department. 

With kind regards,

on behalf of

Dr. Atsushi Asakura 

Academic Editor

PLOS ONE